# Development of Multifunctional Materials Based on Poly(ether ether ketone) with Improved Biological Performances for Dental Applications

**DOI:** 10.3390/ma14041047

**Published:** 2021-02-23

**Authors:** Vanessa Montaño-Machado, Pascale Chevallier, Linda Bonilla-Gameros, Francesco Copes, Chiara Quarta, José de Jesús Kú-Herrera, Florentino Soriano, Victoria Padilla-Gainza, Graciela Morales, Diego Mantovani

**Affiliations:** 1Laboratory for Biomaterials and Bioengineering (CRC-I), Department of Min-Met-Materials Engineering & CHU de Quebec Research Center, Laval University, Quebec City, QC G1V0A6, Canada; vmontano24@gmail.com (V.M.-M.); pascale.chevallier@crchudequebec.ulaval.ca (P.C.); linda-victoria.bonilla-gameros.1@ulaval.ca (L.B.-G.); francesco.copes.1@ulaval.ca (F.C.); chiara.quarta8@gmail.com (C.Q.); 2Centro de Investigación en Química Aplicada, Blvd. Enrique Reyna 140, Saltillo CP 25294, Coah, Mexico; jesus.ku@ciqa.edu.mx (J.d.J.K.-H.); Florentino.Soriano@ciqa.edu.mx (F.S.); victoria.padilla@utrgv.edu (V.P.-G.)

**Keywords:** poly(ether ether ketone), zinc oxide nanoparticles, dental applications, mechanical pro-perties

## Abstract

The main target for the future of materials in dentistry aims to develop dental implants that will have optimal integration with the surrounding tissues, while preventing or avoiding bacterial infections. In this project, poly(ether ether ketone) (PEEK), known for its suitable biocompa-tibility and mechanical properties for dental applications, was loaded with 1, 3, and 5 wt.% ZnO nanoparticles to provide antibacterial properties and improve interaction with cells. Sample cha-racterization by X-ray diffraction (XRD), thermogravimetric analysis (TGA), and differential scanning calorimetry (DSC) as well as mechanical properties showed the presence of the nanoparticles and their effect in PEEK matrices, preserving their relevant properties for dental applications. Al-though, the incorporation of ZnO nanoparticles did not improve the mechanical properties and a slight decrease in the thermal stability of the materials was observed. Hemocompatibility and osteoblasts-like cell viability tests showed improved biological performances when ZnO was present, demonstrating high potential for dental implant applications.

## 1. Introduction

Poly(ether ether ketone) (PEEK) is a relatively inexpensive resin featuring high biocompatibility with bone and versatile mechanical performances suitable for dental applications, mainly due to its elastic module being much lower than current metals used for bone applications. Effectively, current materials used for dental applications, such as titanium and zirconia, present some drawbacks leading to clinical complications due to fracture, corrosion, hypersensitivity, and generation of high stress on the surrounding bone [1,2,3]. Indeed, titanium has an elastic modulus (E-modulus) of 110 GPa [4] and zirconium dioxide has one of 210 GPa [5]. Cortical bone has an elastic modulus of 13.8 GPa and spongy bone has one of 1.38 GPa, as reported in the literature [6]. The difference in E-modulus between bone and biomaterials represents the risk of potential mechanical overloading of the bone, leading to damage to the surrounding tissue. On the other hand, PEEK has an E-modulus of 3–4 GPa [7] close to the natural bone values (cortical and spongy), representing an essential advantage compared with conventional titanium/zirconium-based implants. Furthermore, titanium and its alloys show metal ions release [8] while the market is increasingly asking for metal-free materials [9]. In this context, PEEK represents a valid alternative due to its solid chemistry resistance [10]. Additionally, PEEK has other relevant properties, such as resistance to mechanical and biological degradation and thermal stability (ideal for processes of vapor sterilization), among others [11]. However, currently the main disadvantage of PEEK for dental applications is that it is bioinert, hence limiting osseointegration, which is crucial for the long-term clinical success of implants. Moreover, the behavior of PEEK in the presence of infections, as well as suitable strategies for inhibiting bacterial proliferation in PEEK, have been poorly reported. Indeed, there are only a few studies on the adhesion of bacteria to the resin in pre- and post-operation and they are not comprehensive [12,13]. Taosheng et al., [14] increased the cytocompatibility and osseogenicity of PEEK implants by the incorporation of a multifunctional micro-/nanostructured surface made of hydroxyapatite and nickel hydroxide. In this sense, different types of bioactive agents have been used in order to improve its biological properties [15].

In this context, zinc oxide has been largely studied in the literature for exhibiting intensive ultraviolet absorption and antibacterial properties in the pH interval of 7–8, even in the absence of light. Furthermore, ZnO nanoparticles have been used as filler in many medical materials [16]. Indeed, ZnO has been reported to have selective bacterial toxicity with minimal effect on human cells [17]. The performance of ZnO nanoparticles depends strongly on their shape, size, and surface characteristics. Table 1 describes relevant previous works where the mechanical, antibacterial, and bioactive performance of ZnO-based fillers in polymeric matrices has been reported. Based on the reflected studies, ZnO has a great potential to be used as an antimicrobial and bioactive material for bone cells.

In this project, a multifunctional material based on PEEK and ZnO is proposed for the development of dental implants. The presence of ZnO nanoparticles is expected to provide long-term antibacterial properties to the material [18,19,20,21] while improving its interaction with cells and avoiding thrombus formation upon contact with blood. On the other hand, the preservation—or improvement—of the mechanical and thermal performance of the final composite is also expected regardless of the ZnO concentration.

**Table 1 materials-14-01047-t001:** Previous works of ZnO-based fillers in polymeric matrices showing antibacterial properties and promising interaction with bone cells.

Research	Main Findings	Ref.
**Toxicity of zinc oxide nanoparticles towards bacterial systems**	3.4 mM of ZnO (~13 nm), 100% inhibition of both bacteria *Escherichia coli* (*E. coli*) and *Staphylococcus aureus* (*S. aureus*) was reported.	[17]
**Zinc-modified titanium surface for dental applications**	ZnO nanoparticles have been shown to exert an osteoconductive and osteoinductive effect on mesenchymal cells, where the presence of zinc ions significantly increases the expression of genes related to osteoblasts.	[22]
**Modified Poly (lactic acid) (PLA) and poly(3-hydroxybutirate) (PHB) with ZnO nanoparticles**	High degree of distribution with an acceptable dispersion of the ZnO nanoparticles (~12 nm) in both polymeric matrices, without significant losses in mechanical properties, exhibited excellent antibacterial performance in *E. coli* and *S. aureus* strains for PHB matrix.	[18,19]
**Influence of ZnO nanoparticles on the performance of poly(caprolactone) (PCL) based fibers for wound healing applications**	No PCL/ZnO material (ZnO nanoparticle size around 60 nm) presented cytotoxicity against adult goat fibroblasts and the ability of the composite to heal wounds on the skin of animals without presenting tissue inflammation was shown for a nanoparticle content equal to or less than 4 wt. %.	[23]
**Antibacterial properties of PEEK-ZnO matrices**	Antimicrobial studies of PEEK-ZnO nanocomposites (ZnO nanoparticle size around 60 nm) revealed that the antibacterial activity of PEEK-ZnO nanocomposites against *E. coli* and *S. aureus* was enhanced with ZnO content, and the best antibacterial property was obtained at 7.5 wt. % silane-modified nanoparticles.	[24,25]
**Dual Ag/ZnO-Sulfonated PEEK (SPEEK) with superior antibacterial capability and biocompatibility.**	The antibacterial test showed that Ag-SPEEK and Ag/ZnO-SPEEK effectively inhibited *E. coli* and *S. aureus* reproduction. Additionally, the Ag/ZnO-SPEEK systems promoted an excellent osteoblast cell response evaluated through different tests such as: cell viability, proliferation, spreading, osteo-differentiation, and maturation.	[26]

## 2. Materials and Methods

### 2.1. ZnO Nanoparticles Synthesis 

Zinc acetate (ZnAcO_2_) and potassium hydroxide (KOH) (both from Sigma-Aldrich Química, Toluca, México) were used for the synthesis of ZnO nanoparticles. Methanol (purified by fractional distillation over calcium oxide, CaO) was used as the solvent. The synthesis was carried out in a MARS 6 microwave digestion system (CEM, Charlotte, Matthews, NC, USA) according to the procedure previously reported by Rodríguez-Tobías et al. [27]. Eighteen milliliters of ZnAcO_2_ (0.32 mol·L^−1^) were placed in a polytetrafluoroe-thylene (PTFE) reactor (CEM, Charlotte, Matthews, NC, USA) designed for the microwave equipment. Subsequently, under vigorous stirring, 54 mL of KOH (0.64 mol·L^−1^) were added dropwise. Then, the reactor containing the reaction mixture was placed in the microwave and irradiated for 20 min at 80 °C using a temperature-controlled program. The precipitate obtained was subjected to three cycles of centrifugation (Allegra 64R, Beckman Coulter, Indianapolis, IN, USA) and washing first with water, followed by washing with methanol. The precipitate was finally dried in an oven (Heratherm OGH60, Thermo Fisher Scientific, Waltham, MA, USA) at 70 °C until constant weight. The solid obtained was characterized by X-ray diffraction (XRD, Rigaku, Tokyo, Japan) and images were obtained by both scanning electron miscroscopy (SEM, Jeol, Tokyo, Japan) and transmission electron microscopy (TEM, Fei, Eindhoven, Netherlands). With the help of the micrographs obtained by TEM, the average diameter (Dp) was determined using the Image J program (version 1.48) considering 100–150 particles. 

### 2.2. PEEK-ZnO Matrices Preparation

PEEK (450G from Victrex, Lancashire, England) was dried before being processed for 5 h at 150 °C, according to supplier recommendations [28]. ZnO nanoparticles were dried for 8 h at 100 °C followed by 5 h at 150 °C. To obtain the PEEK-ZnO compounds, PEEK pellets were mechanically mixed with 1, 3, and 5 wt.% ZnO nanoparticles in an Omega 30 Co-Rotating Twin Screw Extruder (STEER Engineering Private Limited, Bangalore, India) under the following temperature profiles: zone 1 200/200 °C, zone 2/3 350/350 °C, zone 4/5/6 400/400/400 °C. The amounts of ZnO used were calculated using 800 g of PEEK as initial material; hence, the amounts of ZnO to prepare 1, 3, and 5 wt.% were 8.08, 24.74, and 42.10 g, respectively.

The filaments obtained were pelletized and 3 mm thick plates (for mechanical and biological tests) were obtained by compression molding under the conditions shown in Table 2.

Once the plates were obtained, they were cut and machined according to ASTM D-638-14. The mechanical properties of the PEEK and PEEK-ZnO composites were carried out on a Universal Testing Machine (Instron 4301 Corp., Wycombe, Bucks, UK); 7 replicates per sample were evaluated. For biological tests, plates were cut and machined for a final diameter of 13.5–14.5 mm.

### 2.3. PEEK-ZnO Matrices Characterization

#### 2.3.1. Crystalline Structure and Thermal Properties

X-ray diffraction (XRD): For the analysis by XRD, an Ultima IV X-ray Diffractometer (Rigaku, Tokyo, Japan) was used with Cu source, operated at 20 kV, 5° per minute, 2θ = 10–80, and 0.02° step width (D-teX Ultra). The crystallinity (*X_c_*) was calculated using Equation (1):(1)$$X_{c}=\frac{\Delta H_{m}}{\Delta H_{m}^{{}^{\circ}}\times X_{PEEK}}\times 100$$
where Δ*H_m_* is the melting enthalpy of the PEEK samples, *X_PEEK_* is the weight fraction of PEEK in the sample, and $\Delta H_{m}^{{}^{\circ}}$ is the melting enthalpy of 100% crystalline PEEK equal to 130 J·g^−1^ [29,30].

The thermal properties were evaluated through thermogravimetric analysis (TGA) Q500, TA Instruments, New Castle, DE, USA) and differential scanning calorimetry (DSC) (Q200, TA Instruments, New Castle, DE, USA). In order to perform the TGA, samples were heated at a heating rate of 10 °C·min^−1^ from 30 to 700 °C and 700 to 800 °C under nitrogen and air atmosphere, respectively. Regarding the DSC, the heating was carried out from 20 to 390 °C at a rate of 10 °C·min^−1^ under a nitrogen atmosphere. Samples were isothermally maintained at 390 °C for 2 min and then cooled down at the same rate to 20 °C. A second heating cycle was conducted under the same conditions.

#### 2.3.2. Sample Preparation before Surface Analysis and Biological Assay

The samples were cleaned using an ultrasonic bath (Elmasonic P, Elma, Singen, Germany) for 10 min in three different solvents in the following order: (1) acetone, (2) deionized water, and (3) methanol. Samples were air-dried after each step and stored in a low-pressure chamber until further analysis.

#### 2.3.3. Dynamic Water Contact Angle (WCA)

The wettability angle of the surfaces was measured by dynamic WCA with a VCA optima XE (AST Products, Billerica, MA, USA). Measurements were performed by adding/removing volume at the material/air interface. A microsyringe was used to place a droplet of 1 μL of deionized water on the surface (advancing angle) with the addition of 1 μL followed by the removal of 1 μL (receding contact angle) to measure the dynamic drop angles. For each sample, 5 drops were placed at several locations on the surface of interest. Reported contact angles are the average of the measurements performed on 2 samples. 

#### 2.3.4. Surface Morphology by Atomic Force Microscopy (AFM)

Surface imaging was performed using an Atomic Force Microscope (Veeco Dimension TM3100, Santa-Barbara, CA, USA) operating in tapping mode with an etched silicon tip (NCHV radius < 10 nm, Bruker, Camarillo, CA, USA). The roughness (rms) was mea-sured over 40 × 40 μm^2^ areas and the results obtained represent the average of three different measurements on one cleaned sample per condition.

#### 2.3.5. Scanning Electron Microscopy (SEM) and Transmission Electron Microscopy (TEM)

SEM images were taken with a JSM 7401-F field emission scanning electron microscope (Jeol, Tokyo, Japan). TEM images were taken with a Titan 80–300 transmission electron microscope (Fei, Eindhoven, Netherlands), and images were collected with a Gatan brand CCD camera of 1024 × 1024 pixels digital resolution.

#### 2.3.6. Cell Viability

The Saos-2 human bone osteosarcoma cell line (ATCC^®^ HTB-85™) used herein was purchased from Cedarlane Corporation (Burlington, ON, Canada). PEEK samples containing different percentages of ZnO (0, 1, 3, and 5%, 5 samples per condition) were cleaned and then sterilized using an autoclave (Crown Steam Group, Fuquay Varina, NC, USA) at 115 °C for 20 min. After the autoclave cycle, samples were put overnight in a 37 °C oven (Fisher Scientific, Ottawa, ON, Canada) to dry completely before further testing. For cell viability tests, human sarcoma osteogenic (Saos-2) cells were cultured in McCoy’s 5A medium supplemented with 15% fetal bovine serum (FBS), penicillin (100 U/mL), and streptomycin (100 U/mL) (all reagents supplied by Gibco, Invitrogen Corporation, Burlington, ON, Canada). Cells were maintained at 37 °C in a saturated atmosphere at 5% CO_2_. Medium was changed every two days until a 90–95% confluence was reached. At this point, cells were detached from the plate using trypsin and then re-plated at a ratio of 1:5. Cells at passage 5 and 6 were used for the experiments. The effect of the different ZnO percentage on cells viability was analyzed using an Alamar Blue-based direct viability assay performed using Saos-2 cells directly seeded on the different PEEK samples (n = 5). Briefly, cells were seeded at a concentration of 20,000 cells/cm^2^ onto the different samples. Culture-treated plastic has been used as a control (CTRL Plast). The different samples were incubated at 37 °C in a saturated atmosphere at 5% CO_2_. After 1, 3, and 6 days, media were removed and cells were incubated for 4 h with a resazurin sodium salt solution. After the incubation, the resorufin product, obtained from the metabolization of the resazurin, was collected and fluorescence intensity at a 545 nm_ex_/590 nm_em_ wavelength was measured using a SpectraMax i3x Multi-Mode Plate Reader (Molecular Devices, San Jose, CA, USA). Data were normalized towards the day 1 CTRL Plast condition and are presented as relative viability ± standard, calculated on mean ± standard deviations of fluorescence readings. Fluorescence intensity is proportional to cell viability.

#### 2.3.7. Clot Formation

In order to study the coagulation time when blood is in contact with the different surfaces, free hemoglobin methodology was performed. Whole human blood (ethic project 2012-815, SCH11-09-091) was collected in citrate-containing blood collection tubes (BD Canada, Missisauga, ON, Canada). Blood donors gave their informed consent for inclusion before they participated in the study. The study was conducted in accordance with the Declaration of Helsinki, and the protocol was approved by the Ethics Committee of Centre Hospitalier Universitaire Québec—Université Laval, Quebec, Canada (FWA00000329, IRB00001242).

Sterile samples were placed in the well of a 12-well multi-plate for the test (1 plate = 1 time point). Briefly, 100 µl of citrated blood were placed on the surfaces of the different samples (3 samples per condition, culture-treated plastic has been used as a control (CTRL Plast)). Immediately after, 20 µL of calcium chloride (CaCl_2,_ Sigma Aldrich, Oakville, ON, Canada_)_) was added to the blood in order to activate the coagulation cascade (CaCl_2_ inactivate the citrate). Samples were then incubated at 37 °C for the selected time points (0, 10, 15, 20, and 40 min). At each time point, 2 mL of deionized water was added to each sample in order to lysate the erythrocyte not entrapped in a blood clot. The aqueous solutions containing the hemoglobin were transferred to 96-well plates to perform absorbance reading. Absorbance at a wavelength of 540 nm was recorded using a Spectra Max i3x (Molecular Devices, San Jose, CA, USA). Absorbance is proportional to the amount of free hemoglobin. Therefore, the higher the absorbance, the higher the amount of hemoglobin and the higher the hemocompatibility. Data were normalized towards the 0-minute results.

#### 2.3.8. Statistical Analysis

For the mechanical test, statistical difference was calculated using ANOVA (Origin 2018) followed by Tukey’s mean comparison. For the viability assay, n = 5 replicates for each condition were used. For the hemocompatibility test, n = 3 replicates for each condition were used. The data shown are means ± standard deviation (SD). Statistical significance was calculated using the ANOVA non-parametric Kruskal–Wallis method with Dunn’s post test through the software InStat 3™ (GraphPad Software, La Jolla, CA, USA). Values of p < 0.05 or less were considered significant.

## 3. Results

### 3.1. ZnO Synthesis

Figure 1 shows some relevant properties of ZnO nanoparticles synthesized in our research group. The X-ray spectra show well-defined peaks at 2θ = 31.8°, 34.4°, 36.2°, 47.6°, 56.6°, 62.9°, 66.4°, and 67.9° that correspond to the diffractions of the crystalline planes (100), (002), (101), (102, (110), (103), (112), and (201), respectively, where the characteristic hexagonal phase of ZnO can be clearly identified. The SEM image (Figure 1b) shows quasi-spherical particles whose average diameter ($\bar{D_{p}}$) was 12 nm, with a diameter distribution slightly ranging from 6 to 20 nm (Figure 1c).

### 3.2. Characterization of PEEK-ZnO Matrices

Figure 2 shows the XRD patterns of ZnO nanoparticles, the diffractogram of the semi-crystalline PEEK, and the corresponding PEEK-ZnO composite materials. Regarding the PEEK matrix, the diffractogram shows the characteristic peaks at 2θ = 18.8°, 20.7°, 22.9°, and 28.9°, corresponding to the diffraction crystalline planes (110), (111), (200), and (211), respectively. In the case of PEEK-ZnO composites, the diffractograms display the diffraction peaks characteristic of both components regardless of the ZnO percentage that indicates the incorporation of ZnO into the PEEK matrix. Additionally, a slight shift to a smaller 2θ angle was observed in the presence of ZnO.

Table 3 shows the summary of the dynamic WCA and roughness (AFM) for the stu-died systems. The WCA was measured in order to study the hydrophobicity of the surfaces. The samples presented an advancing WCA of 78°, 85°, 72°, and 63°, whereas the receding WCA was measured as 33°, 39°, 25°, and 16° for 0, 1, 3, and 5 wt.% ZnO, respectively. As WCA is also dependent on the surface roughness, this property was also evaluated by AFM. The roughness increased from 98 nm for crude PEEK to 130 nm with 1% ZnO and 119 nm with 3% ZnO, whereas the roughness remained unchanged for the sample loaded with 5% ZnO. In the case of samples containing 3 and 5% ZnO, the results clearly exhibit that adding the nanoparticles increased the PEEK hydrophilicity, as expected. Regarding the PEEK-ZnO 1 wt.% sample, the increase in WCA might be a consequence of the significant increase in roughness.

Figure 3 shows the TGA thermograms from PEEK systems with and without ZnO. These thermograms presented two mass losses: the first one corresponds to the polymer degradation and the second one, to the oxidation of residual carbon due to air presence from 700 °C. A slight decrease in thermal stability is observed in the presence of ZnO; specifically, the initial degradation temperature (Td) for the systems at 0, 1, 3, and 5 wt.% ZnO were 562.8, 550.6, 552.0, and 536 °C, respectively. The residues at 750 °C could be attributed to the ZnO content. In this regard, these residues slightly disagreed with the original ZnO concentration (0.46, 2.33, and 3.38 wt.%, for 1, 3, and 5 wt.%, respectively). These results could suggest a non-homogeneous distribution of nanoparticles derived from their tendency to form agglomerates.

Regarding the phase transitions experienced by the PEEK samples, Figure 4 shows the DSC thermograms of the heating–cooling–heating cycle, with the corresponding va-lues of glass transition (T_g_) and melting temperatures. A negligible effect of ZnO nanoparticles on the melting temperature (T_m_) and the glass transition temperature (T_g_) values during the first heating is observed since it is in the order of 341 °C and 166 ± 1 °C, respectively, regardless of the ZnO content. On the other hand, a slight effect in the crystallization temperature (T_c_) and T_m_/crystallinity (Xc) in the cooling and second heating process, respectively, is noticed, and is associated with the presence of ZnO.

Figure 5 presents the TEM images of PEEK-ZnO composites. As it can be observed, when compared to the control PEEK with no nanoparticles (Figure 5a), the images (Figure 5b–d) show the presence of ZnO in different grades of agglomeration, which corresponds to the content of ZnO. A larger agglomerate size is observed with increasing ZnO concentration.

The mechanical properties of the samples have been evaluated and the results are shown in Figure 5. Representative curves *σ* vs. *ε* are presented in Figure 6a; the elastic modulus (E, Figure 6b) is calculated as the slope of the curves *σ* vs. *ε* in the linear elastic region. The mechanical resistance to tension (*σ_max_*, Figure 6c) and the strain at break (*e_break_*, Figure 6d) are also presented. The toughness is calculated as the area under the curve *σ* vs. *ε* (Figure 6e). In Figure 6a, it can be observed that all the tested materials exhibit a linear elastic behavior at *ε* < 8.5%; after that, the beginning of creep or yield is observed. At ε~10%, *σ_max_* is produced followed by a drop in the load capacity of the material. This is the start of plastic deformation, which ends when the fracture is produced. The average values of the elastic modulus (Figure 6b), as well as the *σ_max_*, do not present significant differences between the samples with or without ZnO, regardless of the amount of ZnO incorporated. Figure 5d,e present the comparison of deformation at break (*e_break_*) and toughness, respectively, where the small reduction in values in the presence of Zn nanoparticles is not significant after statistical analysis. 

### 3.3. Biological Performances

The main application envisaged for PEEK-ZnO matrices is dental implants. Hence, it was relevant to study the interaction of materials with osteoblast-like cells in order to predict their osseointegration. To evaluate the effect of the modification of the PEEK material with increasing concentrations of ZnO (1, 3, and 5 wt.%) on cell viability, an Alamar Blue viability assay was performed. As can be seen from the obtained results (Figure 7), a significative reduction is noted against the CTRL Plast condition (1 ± 0.21) for PEEK (0.25 ± 0.0), ZnO 1 wt.% (0.43 ± 0.09), and ZnO 3 wt.% (0.48 ± 0.07) conditions. Moreover, a significative increase can be noted in the presence of the ZnO 5 wt.% (0.58 ± 0.06) condition against the PEEK samples. After 3 days of culture, again, the viability measured for the CTRL Plast (1.59 ± 0.38) condition was significantly higher compared to the PEEK (0.37 ± 0.12) and ZnO 1 wt.% (0.68 ± 0.14). Interestingly, both the viability for the ZnO 3 wt.% (0.88 ± 0.10) and ZnO 5 wt.% (1.04 ± 0.12) resulted significantly increased compared to the PEEK condition. Additionally, the viability for the ZnO 5 wt.% samples is significantly higher than the ZnO 1 wt.% condition. Finally, after 6 days of contact with the different experimental conditions, once again, the viability in the CTRL Plast (2.75 ± 0.67) condition is significantly increased compared to the PEEK (0.62 ± 0.22) and ZnO 1 wt.% (1.19 ± 0.40). As for day 3, with the ZnO 3 wt.% (1.68 ± 0.17) and ZnO 5 wt.% (1.93 ± 0.28) also, a significative increase is noted against the PEEK condition. To be noted, the viability in the pre-sence of the ZnO 5 wt.% samples results being significantly higher compared to the ZnO 1 wt.% condition.

Hemocompatibility results are shown in Figure 8. At the time point of 0 min, as expected, the blood did not coagulate. Moreover, no differences are noted among the different conditions. Therefore, this value has been kept as the reference for the test (100% = maximal hemocompatibility). After 10 and 15 min of incubation, the hemocompatibility of the plastic (CTRL Plast) is significantly reduced compared to the PEEK and 1% and 3% ZnO conditions. Interestingly, the hemocompatibility of the 5% ZnO condition results significantly higher than the 1% ZnO condition at 15 min, with no significant difference at 10 min. At the 20 min time point, again, the PEEK and 1% and 5% ZnO conditions result having a significantly higher hemocompatibility than the plastic condition. Moreover, both the 1% and 5% ZnO conditions show a significantly higher hemocompatibility compared to the PEEK and the 3% ZnO conditions. Finally, after 40 min of incubation, the conditions containing 3% and 5% ZnO result having a higher hemocompatibility compared to the control and PEEK condition.

## 4. Discussion

The synthesis of ZnO nanoparticles rendered a yield of ca. 93 ± 3%. The corresponding diffractogram (Figure 1a) confirms the high purity of the synthesized ZnO nanoparticles; additionally, the presence of slightly wide peaks indicates that the obtained ZnO structure is in the nanometric regime, as confirmed by SEM. Indeed, previous studies showed that when microwave-assisted ZnO synthesis is carried out in aqueous medium, a great variety of morphologies is obtained, predominantly hexagonal structures, whose sizes are far from the nanometric regime [31]. However, in the literature, it is reported that the use of alcohols as a solvent favors the formation of nanometric particles, which has been corroborated by our research group when using methanol. Additionally, various authors [32,33,34] have argued that the difference in solubility of Zn^2+^ ion precursors causes variations in ZnO morphology. However, zinc acetate is less soluble in alcohols than in water, which causes the diffusion of Zn^2+^ ions in solvents such as methanol to be somewhat limited, and in turn, this causes the formation of nuclei (complexes $\mathrm{Zn}{\left(\mathrm{OH}\right)}_{4}^{2-}$) of smaller dimensions (promoting nanoparticle formation) than those generated in an aqueous medium, where the diffusion of ions is stimulated by the high solubility of the precursors. Moreover, it has been demonstrated that ZnO particles on the nanometric scale (<30 nm) and with spherical or quasi-spherical morphologies have a better performance as an antimicrobial agent [35].

Regarding the crystalline structure of the composite materials, the presence of ZnO and PEEK with its characteristic hexagonal wurtzite and orthorhombic unit cell [36], respectively, was evidenced in the diffraction patterns (Figure 2). The narrower peaks in the composites in comparison with those of pure PEEK could be indicative of a decrease in the spherulite’s size promoted by the presence of ZnO, as observed by Jiang et al. [37], as well as a decrease in the crystal domains distribution. The slight shift to a smaller 2θ angle in the presence of ZnO could be indicative of a partial disruption of the crystalline structure of PEEK, provoked by the nanostructures, increasing the *d*-space in the crystalline lattice. Concerning the DSC results (Figure 4), an increase in the ZnO concentration produced a decrease in the melting enthalpy, which is coherent with the XRD results. Moreover, the slight decrease in T_c_ values with the incorporation of ZnO nanoparticles is attributed precisely to the nucleating effect of the nanoparticles, which provokes the crystallization of the PEEK matrix at a lower temperature. However, the slight variations in X_c_ do not significantly affect the toughness of the material, so its mechanical performance for application in the dental field is not compromised.

The PEEK surface shows a hydrophilic behavior with a WCA of 78° (Table 3), which is in accordance with the values reported in the literature [12]. Afterwards, it can be observed that the WCA of PEEK increases up to 85° for PEEK + ZnO 1 wt.%. This behavior can be explained by the increase in the surface roughness. Indeed, PEEK + ZnO 1 wt.% shows the highest WCA and surface roughness. Moreover, it can be observed that the WCA decreases when increasing the incorporation of ZnO nanoparticles in wt.%. This tendency can be attributed to the presence of hydroxyl groups on the surface, which promote the physical adsorption of water molecules through hydrogen–bridge interactions [38,39]. Moreover, the hydrophilic behavior of the PEEK-ZnO surface with roughness on the nanometric scale (Table 3) has been reported as an attractive property for dental applications [40]. Indeed, when comparing the three different PEEK-ZnO matrices, both hemocompatibility and cell viability showed improvements with the increase in ZnO amount (Figure 7 and Figure 8), which correlates with a decrease in WCA.

The thermal stability of the composite systems is slightly reduced by the presence of ZnO. Several authors have reported opposite results [25,37], where the observed behavior is adjudicated to the barrier effect of the well-dispersed ZnO nanoparticles that interfere with the diffusion of the degradation products of the polymer matrix towards the gas phase. Based on the TEM images of the designed systems (Figure 5), the presence of considerable size agglomerates is evident. Therefore, the action of the barrier effect of ZnO is diminished, observing an opposite effect to that reported. However, the PEEK-ZnO system’s thermal stabilities are still good enough for the application.

Concerning the mechanical properties, E, *σ_max_* deformation at break, and toughness are affected by the incorporation of ZnO; these results imply that the ZnO particles are not adequately exploited in the compound, probably due to a poor particle–matrix interaction and the formation of agglomerates as observed by TEM analysis (Figure 5). The slight decrease in toughness and deformation to break reflect a poor interface particle- matrix, which fails early before reaching the maximum deformation of the matrix and a reduction in the deformation energy absorption capacity in the plastic region of the composite materials. Taking these results into account, although an improvement in mecha-nical properties is not observed, contrary to what is reported by other authors [24,37], they remain without significant difference, which does not compromise the performance of the material. However, it becomes evident that the method of incorporating the ZnO nanoparticles and particle–matrix interaction can still be optimized i.e., through ultrasound cavitation of nanoparticles before incorporating them to the PEEK matrix or by modifying ZnO nanoparticles’ surface with silane coupling agents [25,37].

Regarding the biological performance of the samples, a positive effect of the ZnO nanoparticles is observed for both cell metabolic activity and hemocompatibility properties. Indeed, after 6 days of contact of the samples with cells, the highest metabolic activity is observed for the samples with 5% ZnO. The effect of the presence of ZnO nanoparticles on cells has already been studied in the literature, as shown in Table 1. On the other hand, the interaction of PEEK-ZnO with blood has not been previously reported in the literature. In this work, for the first time, a positive effect of decreasing the time of thrombus formation has been evidenced with the addition of ZnO nanoparticles. Indeed, the improvement in hemocompatibility of PEEK is promising for other medical applications such as cardiovascular devices. Several studies have previously tried to improve PEEK hemocompatibility with the grafting of biomolecules as phosphorylcholine-based polymers [41,42] and heparin [43]. The interaction of modified PEEK with plasma proteins has also been studied, which has an effect on the hemocompatibility properties of the polymer. Bartolo et al. studied the interaction of PEEK membranes with albumin, fibrinogen, and immunoglobulin G, concluding that the polymer had low affinity for proteins [44]. Specifically, the low affinity of the polymer with fibrinogen can partly explain its hemocompatibility, mainly because this protein is involved in the coagulation cascade.

## 5. Conclusions

In the present work, matrices of PEEK containing 1, 3, and 5% ZnO nanoparticles were created. The addition of ZnO to the PEEK material, especially at the percentage of 3% and 5%, resulted in increased hemocompatibility and osteoblasts-like cell metabolic activity without a significant impact on the thermal and mechanical properties of the material, showing high potential for dental implant applications.

## Figures and Tables

**Figure 1 materials-14-01047-f001:**
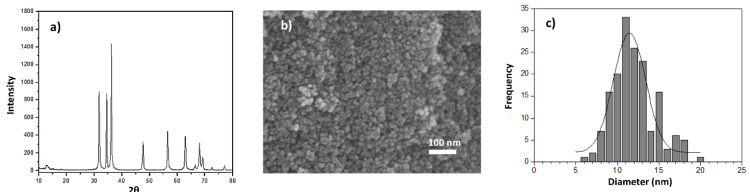
Zinc oxide nanoparticles (**a**) XRD patterns; (**b**) SEM micrograph (scale bar: 100 nm); (**c**) Histogram of ZnO nanoparticles’ diameter distribution.

**Figure 2 materials-14-01047-f002:**
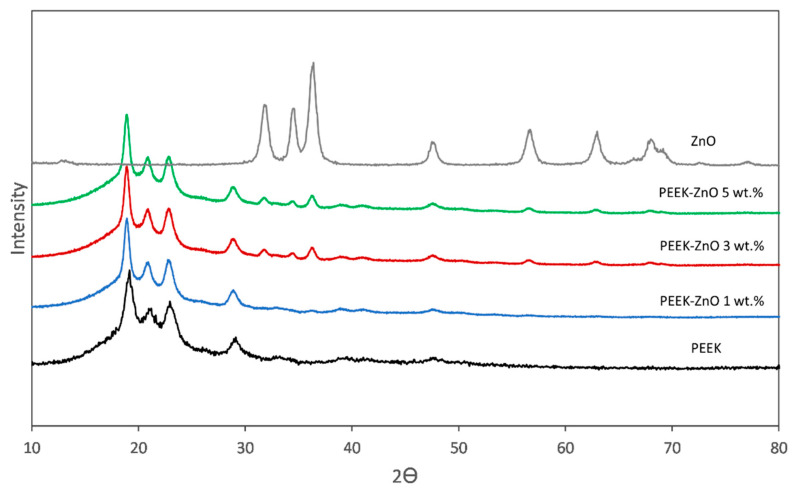
XRD patterns of ZnO nanoparticles, PEEK, PEEK-ZnO 1%, PEEK-ZnO 3%, and PEEK-ZnO 5%.

**Figure 3 materials-14-01047-f003:**
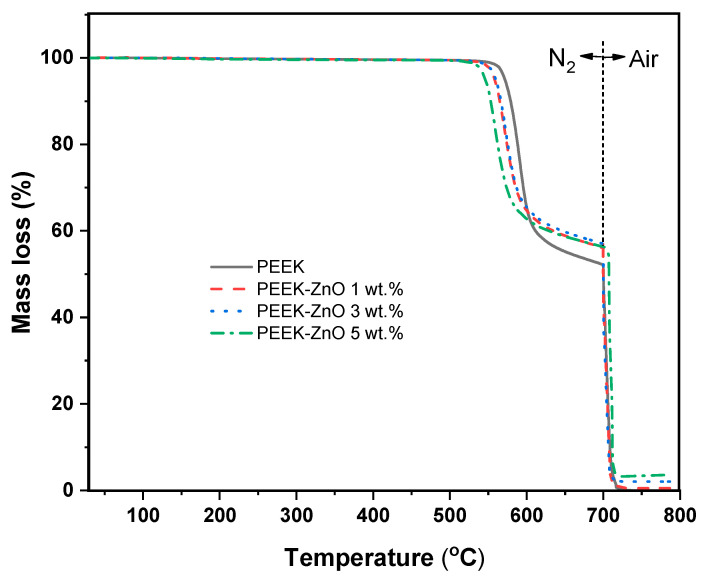
Degradation patterns derived from TGA of PEEK, PEEK-ZnO 1 wt.%, PEEK-ZnO 3 wt.%, and PEEK-ZnO 5 wt.%.

**Figure 4 materials-14-01047-f004:**
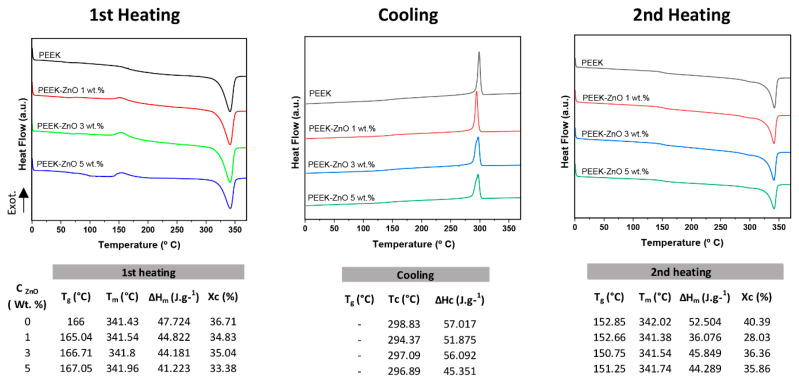
Thermal properties of PEEK, PEEK-ZnO 1 wt.%, PEEK-ZnO 3 wt.%, and PEEK-ZnO 5 wt.%.

**Figure 5 materials-14-01047-f005:**
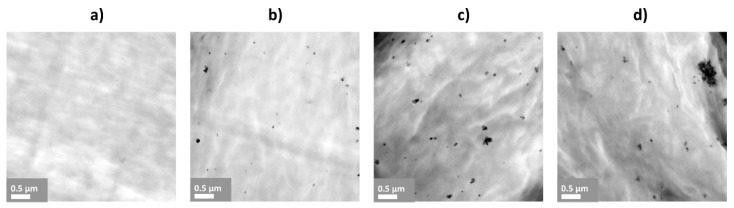
TEM images of PEEK control (**a**) and containing (**b**) 1; (**c**) 3; and (**d**) 5 wt.% ZnO nanoparticles.

**Figure 6 materials-14-01047-f006:**
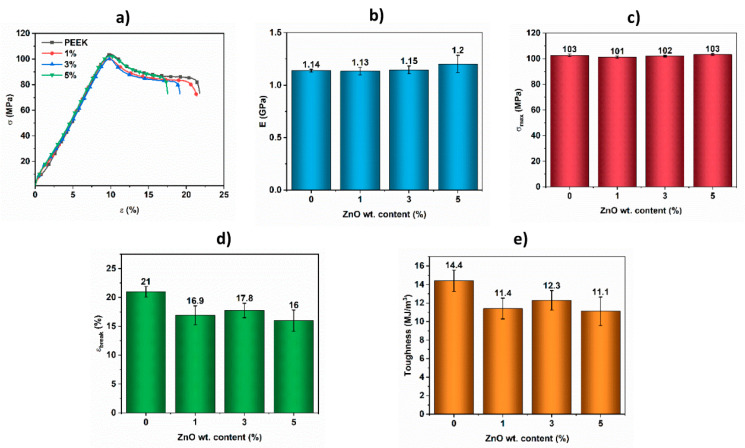
Mechanical properties of PEEK and PEEK-ZnO composites with different ZnO content. (**a**) *σ* vs. *ε* curves; (**b**) E; (**c**) *σ_max_*; (**d**) ε_break_; (**e**) toughness.

**Figure 7 materials-14-01047-f007:**
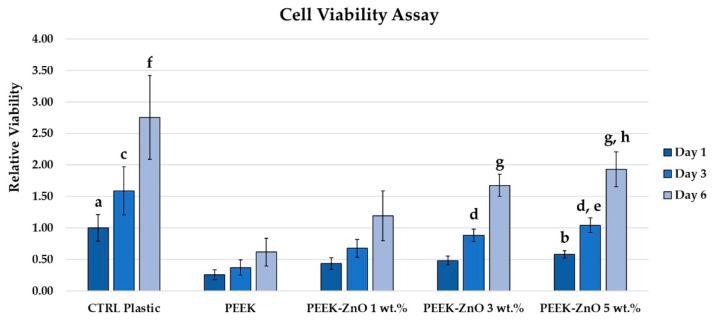
Alamar Blue cell viability assay. (**a**) p < 0.01 vs. day 1 PEEK and ZnO 1% and p < 0.05 vs. Zn0 3%; (**b**) p < 0.01 vs. day 1 PEEK; (**c**) p < 0.001 vs. day 3 PEEK and ZnO 1%; (**d**) p < 0.001 vs. day 3 PEEK; (**e**) p < 0.05 vs. day 3 ZnO 1%; (**f**) p < 0.001 vs. day 6 PEEK and ZnO 1%; (**g**) p < 0.001 vs. day 6 PEEK; (**h**) p < 0.05 vs. day 6 ZnO 1%.

**Figure 8 materials-14-01047-f008:**
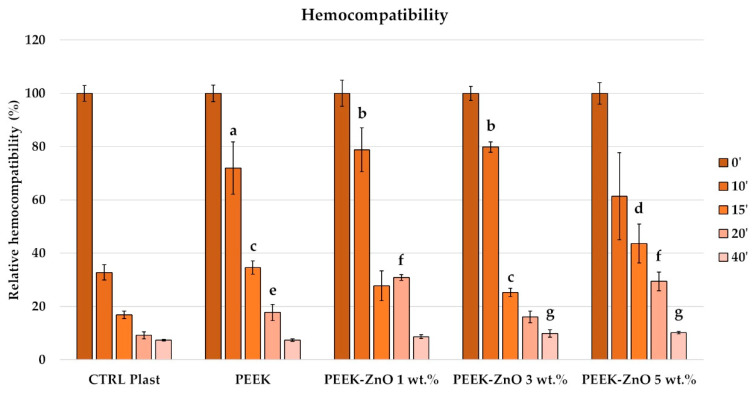
Free hemoglobin hemocompatibility assay. (**a**) p < 0.05 vs. 10′ CTRL Plast; (**b**) p < 0.001 vs. 10′ CTRL Plast; (**c**) p < 0.001 vs. 15′ CTRL Plast; (**d**) p < 0.001 vs. 15′ CTRL Plast and p < 0.05 vs. PEEK 1% ZnO; (**e**) p < 0.05 vs. 20′ CTRL; (**f**) p < 0.05 vs. 20′ PEEK, p < 0.01 vs. 3% ZnO and p < 0.001 vs. CTRL Plast; (**g**) p < 0.001 vs. 40′ CTRL Plast and PEEK.

**Table 2 materials-14-01047-t002:** PEEK and PEEK-ZnO compression plate molding conditions.

Intervals of Time	1°	2°	3°	4°	Cooling Time (s)
Temperature (°C)	370	370	370	370	*
Pressure (Ton)	0	1	10	20	20
Compression time (s)	360	120	60	60	900

***** In order to remove the mold, the plates were left to cool down from 370 to 70 °C. Then, they were pre-heated to 100 °C to avoid thermal shock.

**Table 3 materials-14-01047-t003:** Water contact angle (WCA) and roughness assessed by AFM.

System	WCA (°)		Roughness (rms, nm)
	Advancing	Receding	
**PEEK**	78 ± 3	33 ± 9	98 ± 13
**PEEK-ZnO 1 wt.% **	85 ± 1	39 ± 5	130 ± 8
**PEEK-ZnO 3 wt.% **	72 ± 5	25 ± 4	119 ± 23
**PEEK-ZnO 5 wt.% **	63 ± 7	16 ± 7	98 ± 11

## Data Availability

Data is contained within the article.

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
