# Peer review of "Development of Multifunctional Materials Based on Poly(ether ether ketone) with Improved Biological Performances for Dental Applications"

_materials, 2021, doi:10.3390/ma14041047_

Round 1

Reviewer 1 Report

Dear Author:

The present study was written on a very interesting topic, but some modifications are required.

There is no basis for the determination of the number of specimens in the Materials and Methods section. Provide evidence for the number of specimens.

A conclusion section is needed in this study.

Dest regards

Author Response

Reviewer 1

1.- There is no basis for the determination of the number of specimens in the Materials and Methods section. Provide evidence for the number of specimens.

Answer: thank you for this observation. Most procedures applied in this work have been widely used by the authors. Regarding the PEEK composites fabrication, the parameters stablished were fixed considering data from the provider [1], as well as for the procedure used for PEEK thermal treatment and to the standard methods described in ASTM D-638-14. In the manuscript a couple of lines were added giving more details about the numbers of specimens, please see page 4 lines 5 and 40, page 5 lines 9 and 34.

2.- A conclusion section is needed in this study

Answer: a conclusion was added to the manuscript. Please see page 13, line 34. 

Reference

[1]      Victrex PEEK POLYMER, (n.d.). https://www.victrex.com/en/products/polymers/peek-polymers.

Reviewer 2 Report

1.Introduction

At the end of the introduction, please add the null hypotheses

  1. Materials and Methods

2.1.

Please specify the city and state for the manufacturer Aldrich.

2.2. and 2.3 (with relative subsections of this latter).

Could you please provide a reference for the methodology followed, in order to make data reproducible?

2.3.8.

Please specify the software used for the statistical analysis.

3.

In figure 7 and 8, I suggest adding statistical differences as letters (a, b, c)

Please add references like [1] instead of superscript numbers.

For each figure and table please add the words “Figure ..” and “Table..” in the caption in bold type

Please add a final paragraph for the conclusions

Author Response

1.-Introduction: at the end of the Introduction, please add the null hypotheses.

Answer: Based on the studies developed by other authors, a positive cell-ZnO nanoparticles (ZnO-Nps) interaction was strongly expected. However, the alternative hypotheses regarding the increase of the mechanical and thermal performance of the final composite materials with ZnO-Nps were not as strong as the one previously described, because the later ones have a greater dependence on the preparation method. Therefore, the null hypotheses referring the mechanical and thermal performance prevailed in this work. As was requested, some comments regarding the null hypotheses were added in the introduction, please see page 2 line 25.

2.- Materials and Methods

2.1. Please specify the city and state for the manufacturer Aldrich

Answer: The manufacturer was added to the manuscript. Please, see page 3 lines 3-4.

2.2 and 2.3 (with relative subsections of this latter)

Could you please provide a reference for the methodology followed, in order to make data reproducible

Answer: More details and references regarding the methodology were added. Please see page 3, lines 21-28, page 5 line 43-44.

2.3.8 Please specify the software used for the statical analysis

Answer: The software used for the statical analysis in each case (mechanical and biological studies) are described in page 5 line 51 and 52.

  1. In Fig 7 and 8, I suggest adding statical differences as letters (a, b, c)

Answer: The figures were modified as suggested. See pages 10 and 11.

3- Please add references like [1] instead of superscript numbers. For each figure and table please add the words “Figure..”.. and “Table..” in the caption in bold type

Answer:  All these suggestions were considered and the manuscript was modified.

4- Please add a final paragraph for the conclusions.

Answer: Thank you for this comment. A conclusion was added to the manuscript, page 13 line 34.

Reviewer 3 Report

I am herewith sending comments about materials-1085267 as an attached file.

Reviewer 4 Report

Review of the manuscript “Development of multifunctional materials based on poly(ether ether ketone) with improved biological performances for dental applications.” Overall, it is a well-written and well-organized manuscript and the authors have done a thorough analysis by considering and performing most relevant materials characterization techniques for this study. Besides, this reviewer has some comments and questions that needs to be addressed prior to consideration of this manuscript for publication. Please see below my comments:

  1. The significance of WCA in dental application needs to be explained in the manuscript.
  2. I suggest removing Table 1 and incorporating the details of literature review in the introduction text.
  3. What is the role of the degree of crystallinity of PEEK in characteristics of PEEK-ZnO?
  4. To this reviewer, the major concern is the use tension test for mechanical characterization of the PEEK-ZnO composite. Considering the application and the loading modes in dental implants, indentation seems to be the most relevant mechanical characterization technique. This method has been used for mechanical characterization of dental materials and polymers including PEEK (see references below). I am wondering why the authors have used the uniaxial tension instead of indentation.
  5. Another major concern is not addressing the variation of the fracture toughness of the composites studied in this paper. Please note that the fracture toughness is different from toughness and elongation to break in tension and is measured via studying the crack propagation in solids.
  6. On page 11, the authors have claimed that the WCA is correlated with the surface roughness. In general, this statement is true, however, the data shown in table 3 do not support this claim. For example, PEEK and PEEK-ZnO 5 wt.% have the same roughness value, but exhibit significantly different WCA numbers.

References:

  1. Iqbal, Tanveer, et al. "Nanoindentation response of poly (ether ether ketone) surfaces—A semicrystalline bimodal behavior." Journal of Applied Polymer Science6 (2013): 4401-4409.
  2. Voyiadjis, George Z., Aref Samadi-Dooki, and Leila Malekmotiei. "Nanoindentation of high performance semicrystalline polymers: A case study on PEEK." Polymer Testing61 (2017): 57-64.
  3. He, Li H., and Michael V. Swain. "Nanoindentation derived stress–strain properties of dental materials." Dental materials7 (2007): 814-821.
  4. Drummond, James L. "Nanoindentation of dental composites." Journal of Biomedical Materials Research Part B: Applied Biomaterials (2006): 27-34.

Reviewer 5 Report

This manuscript reported the Development of multifunctional materials based on poly(ether ether ketone) with improved biological performances for dental applications. The biological performances of PEEK were improved by ZnO nanoparticles modification. However, there are some scientific challenges that need to be corrected and answered by the author. The authors should tell the readers the reason and significance of this research. Some specific comments are as follows.

  1. In part 2 materials and methods, please indicate the details of the testing equipment and medicine when it first appears.
  2. The author only gives the mass concentration of ZnO. What is the specific amount of PEEK and ZnO mixed?
  3. 3. In 2.3.2, please correct 10-90.
  4. 4. In Figure 2, the author said “The lower intensity and width of the peaks in comparison with those for pure PEEK could be indicative that ZnO may change the spherulite’s size of PEEK, as it was mentioned by Jiang et al.”. However, the peak strength did not abate, but seemed to increase a little. And, PEEKE-ZnO 5wt. % and PEEKE-ZnO 5wt. % have the same XRD patterns, it doesn't draw your conclusions.
  5. 5. Are there no pictures for WCA and AFM tests?
  6. In Figure 4, Why is the thermal performance of PEEKE-ZnO 1wt. % lower than that of PEEKE-ZnO 3wt. %, and then PEEKE-ZnO 5wt. % lower than that of PEEKE-ZnO 3wt. %
  7. 7. Please keep the full text icon in a consistent format.
  8. 8. In Figure 7, As the amount of ZnO increases, the cell activity increases, can the amount of ZnO also be increased?
  9. 9. Regarding the ZnO killing of bacterial, there are many reports about ZnO based nanomaterials. Compared with the Ag/ZnO nanomaterials (Macromolecular Bioscience, 2018, 18, e1800028), which is the advantages of using only ZnO nanoparticles? The authors should discuss them in the introducniton. In additon, regarding the combination syntheis and application of PEEK with other materials, the introduntion part is not detailed. The authors should read more references (ACS Applied Materials & Interfaces, 2020, 12, 14971-14982) to enrich their knowledge.

Round 2

Reviewer 1 Report

The queries and comments have been responded to as suggested.